# Learning selection cuts with gradients

**Mike Hance⋆ and Juan Robles**

Santa Cruz Institute for Particle Physics, University of California, Santa Cruz

⋆ mhance@ucsc.edu

## Abstract

Many analyses in high-energy physics rely on selection thresholds (cuts) applied to detector, particle, or event properties. Initial cut values can often be guessed from physical intuition, but cut optimization, especially for multiple features, is commonly performed by hand, or skipped entirely in favor of multivariate algorithms like BDTs or neural networks. We revisit this problem, and develop a cut optimization approach based on gradient descent. Cut thresholds are learned as parameters of a network with a simple architecture, and can be tuned to achieve a target signal efficiency through the use of custom loss functions. Contractive terms in the loss can be used to ensure a smooth evolution of cuts as functions of efficiency, particle kinematics, or event features. The method is used to classify events in a search for Supersymmetry, and the performance is compared with common classification tools. An implementation of this approach is available in a public code repository and python package.

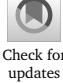
# 1  Introduction

Rule-based data analysis is a common starting point for many studies in high-energy physics. Specific physics processes often motivate phase space requirements, such as cuts on particle momenta or event activity, that are based on physical principles. Optimization of those cut values, using significance or other metrics, can lead to event selections that are easily understood in the context of the physics signature under study. While an analysis based only on one-dimensional cuts will rarely match the performance of a multivariate optimization that fully exploits correlations between particle or event features, the transparency of cut-based procedures, and the relatively straightforward interpretations of the phase space captured by the cuts, make them attractive as a starting point for data analysis. Cuts have been used by many recent ATLAS and CMS searches for supersymmetry to select events for primary signal regions [1–6]. Another use case for cuts is to define "discovery" regions that enable future reinterpretation [7], often by complementing a primary selection based on boosted decision tree (BDT) or neural network outputs.

Cut-based event selections are also useful in other ways. A set of one-dimensional cuts constructs a contiguous volume in feature space that is naturally robust against training samples that do not fully span all features, while deep neural networks identify islands in feature space whose definitions can be more sensitive to the details of the training procedure. This can be especially undesirable in cases when training data are derived from samples of simulated events that may not perfectly model event features used for evaluation. The signal efficiency of a single volume in feature space may be easier to calibrate, which can be useful in measurements where precise knowledge of the signal efficiency is required. For example, the ATLAS experiment uses cuts to discriminate photons from non-prompt backgrounds [8,9]. Cut-based selections are also commonly used in event generators and hardware triggers, where more complicated selections based on neural networks are challenging to implement (though these challenges are quickly being overcome, see [10] for a recent review of machine learning in hardware).

Despite the utility of cut-based approaches, tools to derive optimal cut values are limited, and mostly consist either of TMVA's `kCuts` routine [11], or brute-force scans of feature space (see e.g. the optimization package). TMVA's `kCuts` algorithm is a common starting point for LHC physicists optimizing cut-based analyses; it derives cuts that maximize the background rejection at different signal efficiency working points. Our work was partially motivated by the observation that `kCuts` tunes the cuts for each target efficiency separately, such that small changes in the target efficiency can result in large changes in cuts. Subsets of training events with similar (but not identical) features can also lead to very different cut values. We desired an algorithm that was relatively insensitive to the exact value of the target signal efficiency, and that could provide well-behaved (ideally monotonically-varying) cuts as functions of common event properties like pileup or particle momenta. This also provided an opportunity to implement such an algorithm using modern machine learning libraries.

This work presents an approach to binary classification where optimized cuts are extracted from the parameters of a trained neural network with an extremely simple architecture. Training is performed using gradient descent, requiring selection cuts to be modeled with differentiable functions. The application of a set of selection criteria is modeled as a product of sigmoid-activated linear transformations of input features, where the biases and weights of the linear transforms can be translated into cuts. We refer to this as the "cuts as biases in networks" (CABIN) approach. Custom loss functions are used to derive cuts for specific target efficiencies, and to regulate the evolution of cut values across efficiency targets and other event properties. We test the performance of this technique using a dataset of simulated LHC events, and study the stability of the training procedure for different network hyperparameters.

Ref. [12] studies this problem in the context of cuts used as preselection requirements prior to the use of a neural network classifier, and confronts some of the challenges we discuss here, such as the expression of cuts as differentiable functions. Our work focuses on the optimization of cuts, without including any additional multivariate classifiers, and introduces contractive terms to control the behavior of cuts across working points. However, both Ref. [12] and this work develop methods that are fully differentiable, and can be included as components of larger analysis pipelines that make use of auto-differentiation for training and inference.

## 2 Learning cuts as biases

We consider the problem of assigning binary labels to records $\mathbf{x_i}$ in a data sample, here assumed to be particle collision events. Each $\mathbf{x_i}$ can be represented as a vector of input features: $\mathbf{x_i} = (x_{i,1}, ..., x_{i,k}, ..., x_{i,m})$ where we use $k$ to index the features from 1 to $m$, and $i$ to index the events.

In a typical neural network trained for binary classification, the $m$ features are treated as inputs for a fully-connected feed-forward network that has a single output score $y$, representing a sum of linear transformations of the input values passed through one or more non-linear activation functions, that provides a score for a given event $\mathbf{x_i}$. After training, events from one category will cluster at one end of the network output score distribution, while events from the other category cluster at the other end of the range of scores. A threshold on the output score distribution can then be chosen based on signal efficiency, background rejection, significance, or other metrics. The goal of this work is to derive thresholds on the input features, instead of a single threshold on the output score.

We apply a linear transformation to each input feature $x_{i,k}$:

$$x'_{i,k} = w_k x_{i,k} + b_k \,, \tag{1}$$

where $w_k$ represents a weight, and $b_k$ is the bias. Here we consider a simple case where positive $x'_{i,k}$ correspond to one category, and negative values correspond to another category, with threshold value that separates between the two at $x'_{i,k} = 0$. The threshold in the untransformed space is then:

$$x_{i,k} < -\frac{b_k}{w_k} \,. \tag{2}$$

If the weights are chosen to be unity (up to a sign) then the bias $b_k$ represents the cut value for feature $k$. As a parameter of a linear layer in a differentiable model, the optimal value of the cut can be learned through training.

### 2.1 Differentiable cuts

The problem with learning a set of rectangular cuts, expressed as a product of Heaviside step functions $\Theta_k(x)$, is that step functions cannot be used for gradient-based training. We address this by approximating $\Theta_k(x)$ with the logistic function, $\sigma(x)$, where the inputs are multiplied by some factor $n$:

$$\Theta(x) = \lim_{n \to \infty} \sigma(nx) = \lim_{n \to \infty} \frac{1}{1 - e^{-nx}} \,. \tag{3}$$

This function is differentiable, and for finite $n > 1$ provides a good approximation of the Heaviside function as long as $x$ is not near 0, the threshold used to distinguish between categories. Figure 1 shows the behavior of the logistic function for different values of $n$.

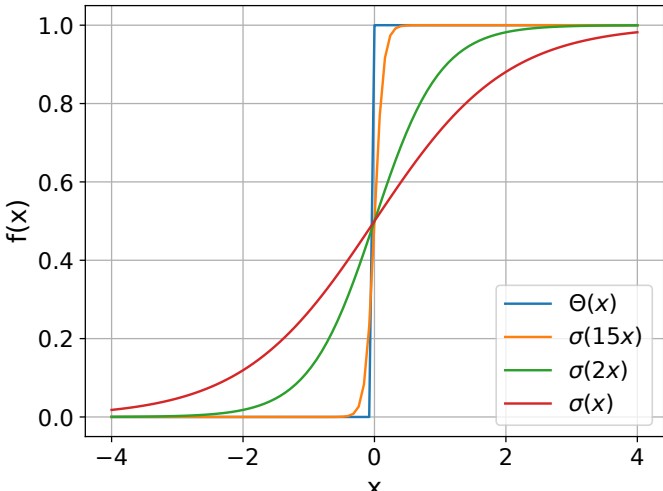

Figure 1: The logistic function $\sigma(x)$ can be used to approximate the Heaviside step function $\Theta(x)$ by scaling the inputs. This provides a differentiable proxy for rectangular cuts.

The result of applying all cuts on features $1..m$ for event $i$ is then calculated as the product of the individual cut functions:

$$\text{score}_i = \prod_{k=0}^{m-1} \sigma(nx'_{i,k}). \tag{4}$$

The score goes to zero if the event fails any cut, and only converges to a value near one if the event passes all cuts (and by a sufficient margin that they are away from the sigmoid turn-on for every input feature).

The final output of this algorithm is a score, not a binary decision. The score asymptotically approaches 0 or 1 for events that fail or pass the selection cuts, respectively, and for sufficiently large values of $n$ the sum of scores for events that are evaluated by the network closely matches the number of events that pass the cuts learned in the training procedure, motivating a large value of $n$. On the other hand, values of $n$ that are too large can make learning unstable, as gradients vanish quickly when $x'$ values move away from the bulk of the training data.

## 2.2 Loss functions

The usual loss function used for two-category classification is binary cross entropy (BCE), which penalizes incorrect classification of any objects in the dataset. Another approach, used by TMVA's `kCuts` algorithm, is to tune the cuts to yield a specified target efficiency, and then maximize the background rejection for that efficiency. In order to reproduce the `kCuts` procedure, which is a common starting point for many HEP analyses, we implement a loss function, referred to here as the "efficiency loss", based on deviations from a target signal efficiency ($\varepsilon_{\text{signal}}^{\text{target}}$), in addition to penalizing non-zero background efficiency:

$$\text{loss}_{\text{efficiency}} = \alpha \left( \varepsilon_{\text{signal}}^{\text{target}} - \varepsilon_{\text{signal}}^{\text{estimated}} \right)^2 + \beta \, \varepsilon_{\text{background}}^{\text{estimated}}, \tag{5}$$

where $\varepsilon_{\text{signal}}^{\text{estimated}}$ is the sum of output scores for true signal events, normalized to the number of true signal events, and $\varepsilon_{\text{background}}^{\text{estimated}}$ is the sum of output scores for true background events, normalized to the number of true background events. Positive-valued hyperparameters $\alpha$ and $\beta$ can be used to control the relative importance of signal efficiency vs background rejection

during training. As discussed in Section 2.2, the choice of $n$ in Equation 4 influences how closely $\varepsilon_{\text{signal}}^{\text{estimated}}$ approximates the actual signal efficiency obtained by applying the cuts derived from the network parameters. Since $\varepsilon_{\text{background}}^{\text{estimated}} \geq 0$, and the term scaled by $\alpha$ is also positive, the loss defined in Equation 5 is convex, and will have a finite global minimum.

Implementation of a custom efficiency loss function also allows the introduction of contractive terms [13] that regulate the behavior of cuts individually and across different optimization regions. The first contractive term we consider is based on the magnitude of the cuts themselves:

$$\text{loss}_{\text{cut size}} = \frac{\gamma}{m} \sum_{k=0}^{m-1} b_k^2, \tag{6}$$

where $b_k$ are the biases, $m$ is the number of features, and $\gamma$ is a positive-valued hyperparameter that controls the influence of this loss term during training. This contractive term provides a penalty based on the magnitude of the cut, and prevents learning cut values that are far from the training distributions.

To enable easy comparisons of the BCE loss with custom loss terms, a BCE loss term, scaled with parameter $\delta$, is also added to the efficiency loss. The $\delta$ parameter can be set to small values (or zero) to ignore the BCE term, or can be set to one with all other loss hyperparameters set to zero to produce exactly the BCE loss.

# 3 Data

We assess the CABIN approach using simulated collider physics data, focusing on the problem of discriminating events arising from Supersymmetric interactions from those of the Standard Model (SM). In particular, we simulate the production of pairs of sleptons, scalar SUSY partners of SM leptons, which are separated in mass from the lightest neutralino, here also the lightest SUSY partner, by approximately 20 GeV. The final state includes two opposite-sign leptons from the slepton decays, along with missing transverse energy ($E_{\text{T}}^{\text{miss}}$) from the invisible neutralinos. In such scenarios, it is often advantageous to require a hard jet, nominally from initial state radiation, which boosts the SUSY system. The SM production of $WW$ events, with both bosons decaying leptonically, provides an irreducible background that is difficult to suppress. Example Feynman diagrams of the SUSY signal and irreducible SM background process is shown in Fig. 2. The ATLAS collaboration recently studied this final state in Ref. [7], where it used cut-based selections to test for generic excesses as well as BDT-based selections to constrain specific Supersymmetric scenarios.

We generate events for signal and background processes using the MAPYDE toolkit [14], with MADGRAPH5_AMCNLO version 5.2.7.3 [15] and NNPDF 2.3 leading order parton distribution functions [16]. Events are showered with PYTHIA version 8.244 [17], and passed through a parameterized simulation of the ATLAS detector response implemented in DELPHES version 3.5.0 [18–20]. Events in the SUSY search are pre-selected to have two leptons with $p_{\text{T}} > 10$ GeV, at least one energetic jet ($p_{\text{T}} > 100$ GeV), and substantial missing transverse energy ($E_{\text{T}}^{\text{miss}} > 300$ GeV). The generated samples have 88k signal events and 33k background events after preselection, 80% of which are used for training, and 20% of which are used for testing.

Training procedures for all methods use the following event features:

- $E_{\text{T}}^{\text{miss}}$,

- the $p_{\text{T}}$ of the leading jet,

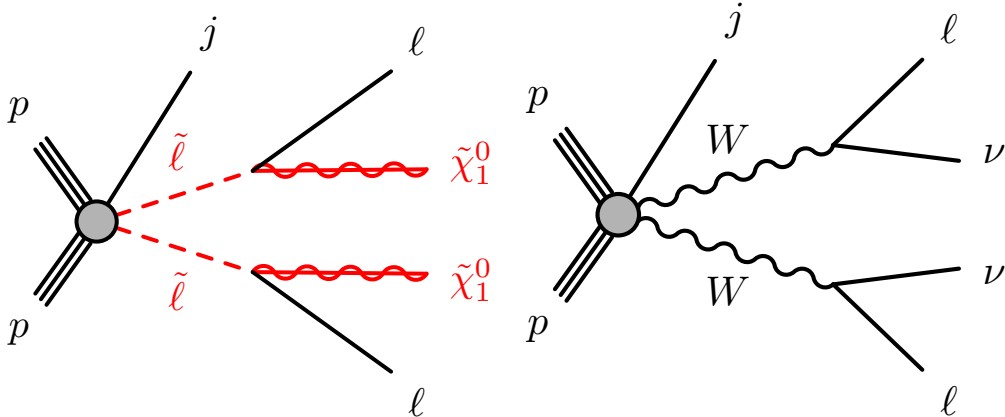

Figure 2: Feynman diagrams for SUSY slepton production (left) and background from SM diboson production (right).

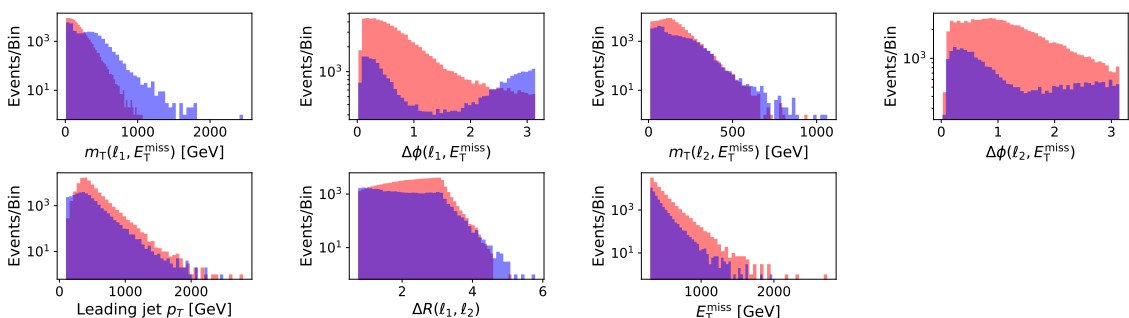

Figure 3: Distributions of kinematic features used to classify SUSY and SM $WW$ events.

- the separation in $\phi$ between each lepton and the missing transverse momentum: $\Delta\phi(\ell, E_T^{\text{miss}})$,

- the transverse mass of each lepton and the $E_T^{\text{miss}}$:

$$m_T = \sqrt{2E_T^{\text{miss}} p_T(\ell)\left(1 - \cos\Delta\phi\left(\ell, E_T^{\text{miss}}\right)\right)},$$

- the separation $\Delta R$ of the leptons from each other:

$$\Delta R = \sqrt{(\Delta\phi(\ell_1, \ell_2))^2 + (\Delta\eta(\ell_1, \ell_2))^2}.$$

# 4 Implementation

The CABIN method is tested using the SUSY dataset described above, and compared with three alternative approaches: a cut-based selection tuned using TMVA's `kCuts` algorithm; a simple, fully-connected neural network; and a BDT.

## 4.1 CABIN method

We implement the CABIN method in the `cabin` python package [21] using `pytorch` [22]. The CABIN method optimizes a sequence of orthongonal 1-dimensional cuts to partition the

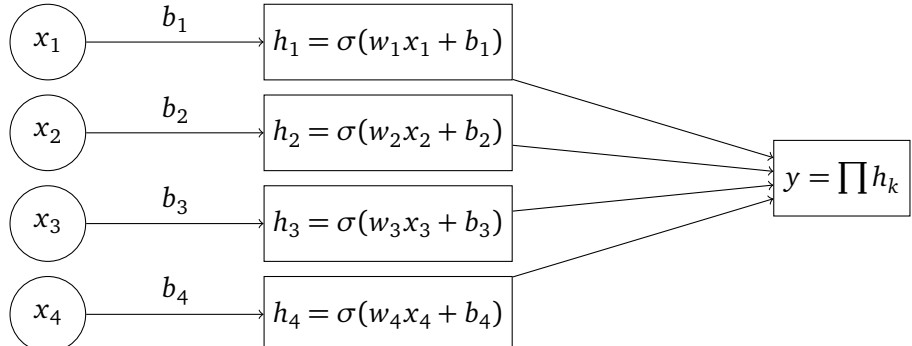

Figure 4: A one-to-one linear network used to biases ($b_k$) that correspond to cuts. The sign of the weight ($w_k$) determines whether the bias term corresponds to a "greater-than" or "less-than" cut; for simplicity we usually fix them to $\pm 1$.

$m$-dimensional feature space into signal-enriched and background-enriched regions.

A one-to-one linear network is constructed in which $m$ features ($m = 7$ in the SUSY dataset) are inputs to $m$ nodes in a single hidden layer, with each input connected to one and only one hidden node. The network is illustrated in Fig. 4. The input feature first undergoes the linear transformation in equation 1, followed by a logistic activation function. The outputs of the hidden layer are multiplied to produce a single output score for each event:

$$\text{score}_i = \prod_{k=0}^{m-1} \sigma\left(n\left(w_k x_{i,k} + b_k\right)\right). \tag{7}$$

We find that $n = 15$ results in output scores for signal events that are close enough to unity that the sum of the output scores is a good estimate of the number of signal events that pass the derived cuts, while still avoiding vanishing gradients during optimization. The weights $w_k$ can either be fixed by the user, or learned as part of the training procedure. We prefer setups in which the weights are fixed to $\pm 1$, such that they only encode whether a cut represents a "greater than" or "less than" requirement.

The training procedure is sensitive to the initial values assigned to the weights and biases. If the initial weights and biases result in cuts that are far from the bulk of the distribution then the gradients of the loss function can vanish, leading to poor training convergence. Initializing cut values in the middle of the distribution helps with this. We use the `StandardScaler` function in `scikit-learn` [23] to rescale the inputs to have zero mean and unit width. The biases are then initialized to 0, which helps avoid vanishing gradients at the start of training.

CABIN networks are trained with the `torch.optim.SGD` optimizer with a learning rate of 0.5, processing data in a single batch for 100 to 200 epochs. Since these networks are simple by construction, training times are typically fast, and no attempts were made to optimize the training hyperparameters or tune for fast convergence of the loss. The hyperparameters included in the loss functions are summarized in Table 1, along with values used during training. CABIN networks are studied with both a pure-BCE loss function as well as a loss that targets a specific signal efficiency.

### 4.1.1 Efficiency scan networks

The one-to-one linear networks described above, in combination with the custom loss function that trains a network to yield a target signal efficiency, allows the construction of a collection of simultaneously-trained networks that span multiple efficiency targets. The loss function for this collection of networks incorporates efficiency loss terms for each of the sub-networks,

Table 1: Hyperparameters used in the custom loss functions. In studies of the BCE loss, $\delta$ is set to 1.0 and all other terms are set to 0. When deriving cuts for target efficiencies, $\delta$ can be zero or small compared to $\alpha$, $\beta$, and $\epsilon$.

| Parameter | Use | Typical value |
|---|---|---|
| $\alpha$ | Importance of reaching target signal efficiency | 1e+1 |
| $\beta$ | Importance of minimizing background | 1e-1 |
| $\gamma$ | Overall magnitude of cuts | 1e-5 |
| $\delta$ | BCE term | 1e-3 |
| $\varepsilon$ | Smoothness of cuts across signal efficiencies | 1e0 |

as well as contractive terms that regulate the behavior of the cut values across different sub-networks. The contractive smoothing terms are implemented as penalties for cuts that are not between cuts for neighboring target efficiencies:

$$\text{loss}_{\text{smooth}} = \varepsilon \sum_{k=1}^{m-2} \frac{\left(b - \bar{b}\right)^2}{(\Delta b)^2 + \rho}, \tag{8}$$

where $b$ is the bias (cut) for a given target efficiency, $\bar{b} = \frac{b_+ + b_-}{2}$ is the mean of the biases for adjacent higher ($b^+$) and lower ($b^-$) target efficiencies, respectively, and $\Delta b = b^+ - b^-$ is the range between adjacent cut values. This loss term encourages the cut to be between adjacent cuts, while $\rho$, which we set to 0.1, avoids divergences in cases where the cuts for adjacent target efficiencies are identical (e.g. at the beginning of training).

The efficiency scan networks are implemented as a collection of one-to-one linear networks correlated through the $\varepsilon$ term in Table 1. The efficiency scan networks implemented here have target efficiencies ranging from 10% to 90% in steps of 10%, as well as targets at 1%, 5%, 93%, and 95% to explore the behavior at more extreme values of the target signal efficiency.

## 4.2 Simple neural network

For comparison, we also implement a simple version of a neural network, with $m$ inputs, one output, and no hidden layers, illustrated in Figure 5. In this approach the $m$-dimensional feature space is partitioned with a hyperplane with optimized orientation in $m-1$ dimensions. The network output is calculated as:

$$\text{score}_i = \sigma\left(\sum_{k=0}^{m-1} w_k x_{i,k}\right). \tag{9}$$

This network, with trainable weights and no biases, has the same number of free parameters ($m$) as the one-to-one linear network in equation 7 that has trainable biases and fixed weights. A BCE loss function is used with this model. The learning rate was set to 0.1, compared to 0.5 in the CABIN networks, to avoid oscillations in the loss.

## 4.3 TMVA `kCuts`

The TMVA `kCuts` method is also used to train cuts for the example datasets. Signal and background events have unit weights in the training. The `kCuts` routine can be configured to use MC sampling, a Genetic Algorithm, or Simulated Annealing. The Genetic Algorithm is described in the manual [11] as having the best performance, and is used here. The training time for `kCuts` varies linearly with the product of the population size, number of steps, and

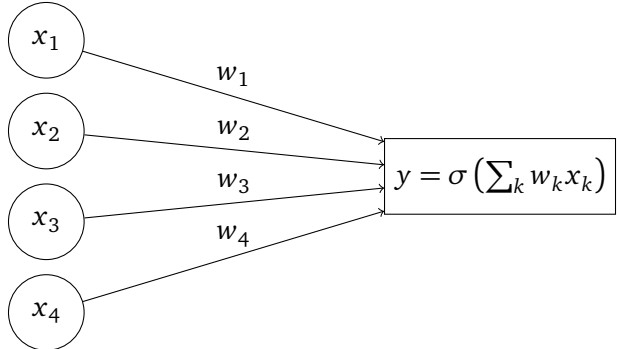

Figure 5: A simple linear network, with trainable weights and no bias terms.

number of cycles (roughly corresponding to the product of batches and epochs when training a neural network). We tested two configurations:

1. population = 200; steps = 10; cycles = 5,

2. population = 500; steps = 20; cycles = 6.

The behaviors of the output cuts and ROC curves of the two approaches are qualitatively similar. We use configuration #2 for comparison with the CABIN approach. We also use the `EffSel` method to target specific efficiencies, the `VarProp=FSmart` option to categorize cuts, and employ the `CreateMVAPdfs` option to rescale the data for processing internally.

### 4.4 Boosted decision tree

Finally, a BDT is trained to provide a ceiling for the performance of cut-based approaches. We use the `xgBoost` [24] toolkit for training a BDT to separate signal from background. BDT's with a `binary:logistic` objective function and a `logloss` metric were trained with a range of estimators (500 through 2000, in steps of 500) and depths (2 through 7, in steps of 1) and tested for convergence, accuracy within a test sample, and overtraining through comparisons of training and test losses. Signs of overtraining were observed for depths beyond 3, and overtraining was exacerbated for large numbers of estimators. A configuration of 500 estimators and a depth of 3 was used for comparisons with the CABIN approach.

## 5 Results

The ROC curves for the methods above applied to the SUSY data sample are shown in Fig. 6. The BDT provides the best performance as measured by the summary AUC scores. The CABIN efficiency scan, TMVA `kCuts`, and simple fully-connected neural network all achieve equivalent AUC scores, though the ROC curves demonstrate that the methods have different performance depending on the target efficiency. The performance of a set of cuts derived using a one-to-one linear network with a BCE loss function are also shown for comparison.

The optimized cuts for the SUSY dataset from the efficiency scan network, from TMVA `kCuts`, and from a one-to-one linear network trained with a BCE loss function are shown in Fig. 7. The efficiency scan network succeeds at ensuring a smooth (but still nonlinear) evolution of cut values for the individual input features in the SUSY dataset, while the cuts tuned by TMVA `kCuts` vary significantly as a function of the target efficiency.

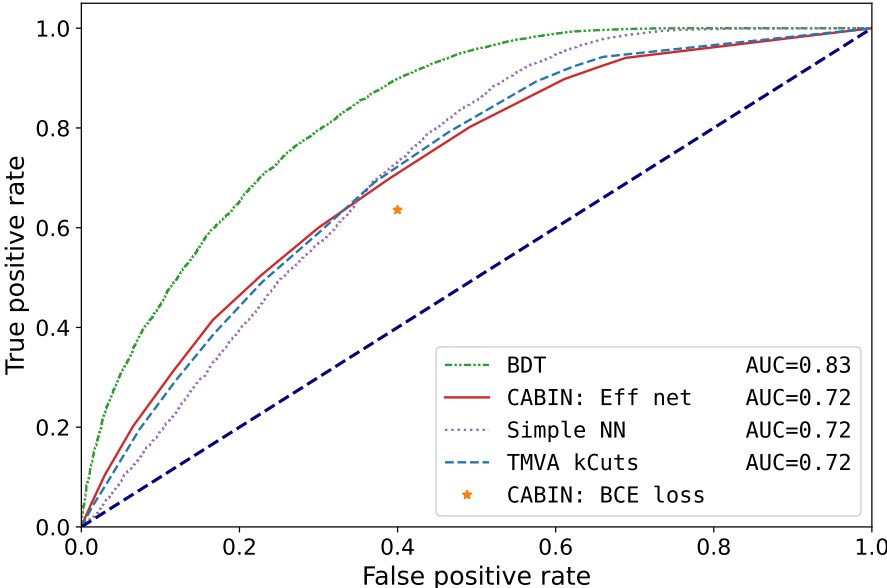

Figure 6: Summary of ROC curves and AUC scores for different methods applied to the SUSY dataset. For the CABIN and TMVA methods, true-positive and false-negative rates are assessed after applying the cuts derived from those methods. ROC curves for the BDT and Simple Neural Network methods are derived from the distribution of output scores.

The loss distribution for the efficiency scan network trained on the SUSY dataset is shown in Fig. 8, including the contributions from the different terms in the training loss. The loss terms associated with reaching the desired signal efficiency and minimizing the background efficiency dominate the overall loss, but the smaller terms associated with the size of the cuts and the smoothness of cuts across target efficiencies regulate the behavior and prevent the cuts from settling into inappropriately loose or tight values. The impact of the BCE term appears large, but removing it has no impact on the derived cuts.

## 6 Discussion

Event selections using pass/fail cuts on input features are ubiquitous in high-energy physics, from phase space restrictions in Monte Carlo Event generators to hardware triggers and offline analysis. Despite their prevalence, cut optimization is often left to *ad hoc* procedures, as opposed to more optimization of more complicated discriminants based on neural network or BDT outouts.

To fill this gap in optimization tools, we present the CABIN approach to learning optimized cuts for event classification, where the cut values are represented by biases in a type of linear neural network. The CABIN technique uses gradient-based learning and is aware of useful HEP metrics such as signal and background efficiency. The method can self-regulate cuts across working points to deliver both good performance and robustness against statistical noise. An implementation of this approach is available in the `cabin` python package [21].

Extensions of this approach can be used to optimize cuts based on significance metrics [25], possibly with additional requirements to ensure signal yield large enough to enable discovery or limit setting. Another foreseen extension will be ensuring the smooth evolution cuts as functions of particle or event properties that cause correlated changes in input features. For

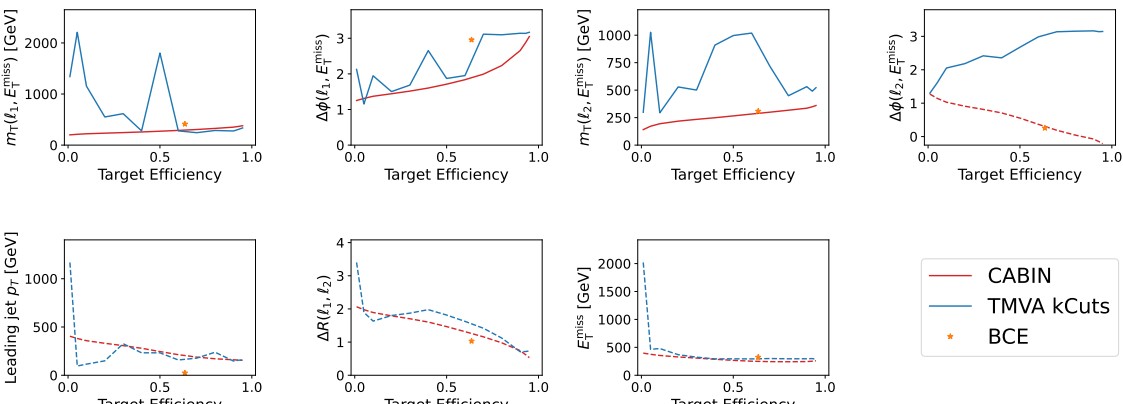

Figure 7: Cuts on SUSY dataset features extracted from the trained one-to-one linear networks in the efficiency scan ensemble, and from TMVA `kCuts`, as a function of the target efficiency. The cuts from the BCE-trained network are also shown for comparison. Solid lines indicate "greater-than" cuts, while dashed lines indicate "less-than" cuts. The cut on $\Delta\phi(\ell_2, E_T^{miss})$ is the only one where `kCuts` and CABIN have opposite orientation.

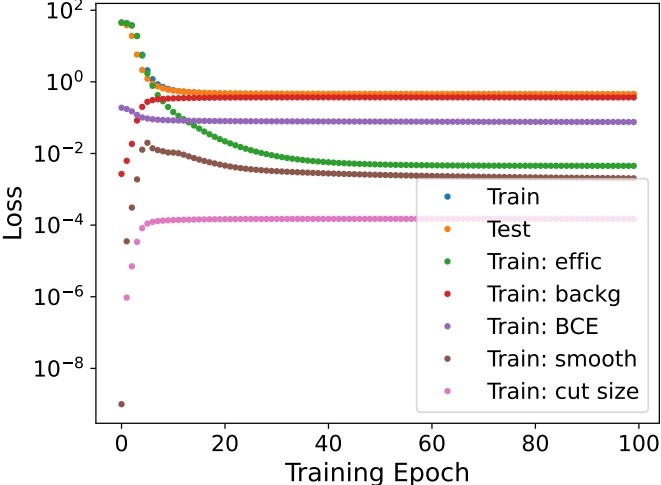

Figure 8: The loss distribution for the efficiency scan network, for training and test subsets of the SUSY dataset. The loss is composed of different terms, corresponding to the signal efficiency, background efficiency, cut sizes, and smoothness conditions, along with a BCE term, all scaled by independent hyperparameters listed in Table 1.

example, when training a classifier for particle identification, the input features may vary with particle energy, and will likely evolve monotonically, though possibly in different ways for signal and background. Contractive terms in the loss function can ensure that cuts on input features for neighboring energy bins have similar values, consistent with the expected smooth evolution of the features themselves.

Finally, as suggested in Ref. [12], a CABIN network can be embedded as one step in a chain of differentiable analysis selections. This provides a path towards full-chain optimization of cut-based event generation, triggering, and analysis pre-selection, followed by final optimization using fully connected networks, all within the same minimization task.

# Acknowledgments

We would like to thank Earl Almazan, Scott Phillips, and Giordon Stark for helpful discussions on the `cabin` approach.

**Funding information** This work was supported by the U.S. Department of Energy (DOE) Office of High Energy Physics under Grant No. DE-SC0010107.

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
