# Peer review of "Learning Selection Cuts With Gradients"

_SciPost Physics Core, doi:SciPost Phys. Core 8, 079 (2025)_

## Round 1 · Referee Report · Matthew James Stephenson (Referee 1) · 2025-2-26

Strengths

  1. preselection optimization: creating robust event filters before multivariate analysis.

  2. trigger system design: physical intuition of cut-based analyses

  3. systematic uncertainty quantification: propagating errors through differentiable cut parameters

Weaknesses

The ATLAS_significance_loss function is a notable weakness because:

1) division-by-zero risk: when background (b) approaches zero, the calculation n * torch.log(n/b) becomes numerically unstable

2) incorrect sigma handling: the sigma=0 approximation fails to account for vanishing background scenarios

a corrected implementation is posted here as a pull request at the publicly available supplemental github repo at this submission which fixes this by adding value clamping for logarithmic arguments, protects against negative values in physical quantities and maintains differentiability through masked tensor operations and is available to merge at pull request https://github.com/scipp-atlas/cabin/pull/6

Report

arXiv:2502.08615:

numerical stability is of critical importance at machine learning applications for physical analysis. 2502.08615v1's effectiveness for cut optimization 'should' properly extend to low-statistics regimes and pure signal scenarios if and only if it's implementation is corrected by approving the proposed PR (formatting issues, should they arise can be corrected post-merge) at the publicly accessible repository paired to this submission at https://github.com/scipp-atlas/cabin/pull/6

mathematical demonstration of critical error
For pure signal samples where b → 0:

n = s + b ≈ s
x = n * log(n/b) → s * log(s/0) → ∞
This produces infinite loss values and NaN gradients during training.

-Matthew James Stephenson

Requested changes

critical implementation errors were located at https://arxiv.org/abs/2502.08615v1's significance calculation (at accessory data hosted at github) and updated code was provided as a pull request to resolve the numerical instability issues at https://github.com/scipp-atlas/cabin/pull/6

Recommendation

Ask for minor revision

  • validity: ok
  • significance: good
  • originality: good
  • clarity: good
  • formatting: good
  • grammar: good

Author:  Michael Hance  on 2025-03-04  [id 5266]

(in reply to Report 1 by Matthew James Stephenson on 2025-02-26)

Hello,

Thanks for these comments on the code and the pull request to address the issues that were raised. The PR has been accepted and merged into the main branch of the github repo. We do not see that these comments motivate any changes to the paper itself, but the improvements to the code were welcome.

Best wishes,

-Mike

---

## Round 1 · Referee Report · Wouter Verkerke (Referee 2) · 2025-5-28

Strengths

1- The paper presents a nice overview of methods to learn optimal selection cuts for a relatively straightforward scenario - selection only based on single cuts observables, and studies and compares the behavior of several approaches. It is an interesting optimisation problem that is not traditionally studied.

2- The methodology and procedural aspects in Sections 2 and 3 are generally well explained I have no comments on these

3- The description of the numeric behavior of the training with various target goals is well presented in paper, with insightful comments that explain the behaviour in various cases

Weaknesses

1- I am some reservations of on the use of the studied selection model in real-world particle physics examples - where in the cut-based approach boosted decision trees have been the standard since at least a decade as opposed to single cuts. The mention of one or more known real-world examples where the simple cut based has actually been preferred over BDT-style approaches would strengthen the introduction of the article.

2- My main comments are on Section 5, where I am rather puzzled by the choices of the samples. The Gaussian sample effectively represents a multi-variate Gaussian distribution for both signal and background. The comment that means and widths are randomly chosen is confusing - this could mean many things, but reading the entire section I conclude that you mean that the chosen values are arbitrary (but not random in the sense that they differ from event to event, or from sample to sample). I suggest that this is more clearly worded. The choice of arbitrary values is however puzzling - how is this meant to be interpretable benchmark?

3- The paper would generally benefit from a description on why this is a good bench mark, and provide a rationale for its choice (not for every parameter, but in general how ‘easy’ or ‘difficult’ of a problem it represents). The fact that you later observe in Section 6 that AUC values of 99% of the ROC curves are achievable is surely a function of chose of means and widths you have ‘arbitrarily’ made, which makes it difficult to draw conclusions from this I think?

Report

(Some general comments)

a- In Section 4 two networks are introduced, ‘one-on-one-linear’ and ‘simple linear’, which are described in mathematical terms, but in effect correspond to ‘a sequence of 1D cuts’ vs a linear decision plane with an optimal orientation in the nD feature space, which make it easier to visualize/understand in my mind what the difference approaches entail. I would suggest to add such more intuitive descriptions to this section to better explain the models.

b - On the description of the Gaussian model: since the selection problem in each feature dimension is invariant under linear rescaling, the chosen parameterization is also redundant: either signal or background can always be chosen as a unit Gaussian without loss of generality. Why not do that?

c - The other sample, a SUSY sample, represents rather the other extreme, with non-Gaussian and strongly correlated distributions in ways that are not easy to understand. While the performance is indeed easily seen to be very different, it is difficult to draw hard conclusions from this extreme comparison. Why not add an intermediate scenario of multi-variate Gaussians with some degree of correlation, which the simple linear model is better equipped to adapt to than the one-to-one model by virtue of its design?

d- Also - referring back to my opening comment - the SUSY model chosen here — is of such complexity that it would not be natural target for a ‘single observable cuts’ strategy in a real-world analysis, so a brief explanation on why this is relevant test case would be a helpful addition to the paper.

e- Section 6.1 - The observed excellent performance is a result of the choices you have made for the means and widths. Is
the chosen benchmark, where near-perfect results are obtained with a very simple model the most relevant benchmark for the cabin procedure benchmarking? Or would have a more dataset a more challenging discrimination task been more insightful to do this benchmark. It would be nice if the paper could give a bit more insight on this.

f - Discussion - I would argue that the choice of pass/fail cuts in MC generation and trigger is traditionally intentionally not chosen as the result of an numerical optimization, but is rather based on physics insights or knowledge of down-stream analysis strategies, with the goal to minimize the impact there. The most promising motivation I only see at the end, where it is presented in the context of differential programming, perhaps it is worthwhile to lift this motivation for this work to the introduction, rather than mentioning it at the very end?

(Report)

Overall I assess this paper to meet the criteria for the journal, but a few clarifications in some places would be needed (but only amount other a minor revision IMHO)

Requested changes

1- An improved discussion of actual use cases for the ‘series of single cuts’ strategy in the introduction would strengthen the paper (the current example like phase space cuts in generation are typically the ones where values are not chosen with an optimization strategy, but are based on other insights.) If possible some concrete example (with refs) would be nice, but is not not essential. I would certainly suggest to mention the future use case in differential analyses here.

2- A better motivation in Section 5 for the choice of both samples in the study, in particular for the chosen configuration of the Gaussian model, which seems arbitrarily configured. Why is this is a relevant benchmark, and why is the chosen configuraion where very high separation can be achieved with simple cuts an insightful benchmark scenario. The same for the SUSY model, which is of such complexity that is hard to imagine that ML optimized single-cuts would every be used in an actual analysis.

Recommendation

Ask for minor revision

---

## Round 1 · Referee Report · Anonymous (Referee 3) · 2025-6-15

Strengths

1) The paper is well written and describe the methodology in adequate details 2) The paper provides an orginal idea, making a classification with a set of one-dimensional cuts which are selected using a neural network

Weaknesses

1) The method should be compared to other alternative methods. For example, no comparison with boosted decision tree, a popular method, is done 2) It isnot clear that this method provides a more robust solution against the fluctuations in training sample. No proof or study is perfomed on this 3) The gaussian comparison is rather weak , since it is a too trivial example, where it is easy to get an optimal classifier 4) When comparing with the SUSY sample, it is not clear what is the statistics used, which type of neural network is used (was some hyper-parameter optimisation done in such case, was some preprocessing perfomed for the initial variables, is some regulariation, dropout or similar using during training,....). Here would be nice to see also a comparison with BDT. 5) It is also missing a comparison with the TMVA cut method, showing this is superior

Report

I think the paper is worth publishing, but it would need a more detailed study on the advantages of the proposed method. In addition to what mentioned above, I think it would be good to study the robustness of the method using independent training data, showing is more stable (having a reduced variance, and maybe bias) compared to alternative methods (BDT, NN)

Recommendation

Ask for minor revision

---

## Round 2 · Referee Report · Anonymous (Referee 3) · 2025-9-30

Report

Thank you for carefully addressing the comments raised in the original version of the paper. I particularly appreciate the inclusion of the comparison with the BDT method and to focus the comparison on a more realistic dataset. I have no further remarks, and in my view the paper is ready for publication.

Recommendation

Publish (meets expectations and criteria for this Journal)

---

## Round 2 · Referee Report · Wouter Verkerke (Referee 2) · 2025-10-1

Report

Thank you for this revised version of the paper. I think it reads well now, and have have no further comments (except for one small suggestion below)

Requested changes

Figure 7 shows a rather striking difference in cut stability vs efficiency for TMVA and CABIN (with the latter being more stable), you observe this also in your results section, but a short sentence on why this is the case would be a nice addition (if possible).

Recommendation

Publish (meets expectations and criteria for this Journal)

  • validity: -
  • significance: -
  • originality: -
  • clarity: -
  • formatting: -
  • grammar: -

Author:  Michael Hance  on 2025-10-13  [id 5920]

(in reply to Report 2 by Wouter Verkerke on 2025-10-01)
Category:
answer to question

Thank you for your comments! We have added a short sentence in the paragraph on the plot of the loss values, noting that the "smoothness" term in the loss is responsible for the stability of the CABIN cuts compared to TMVA.

---

## Round 2 · Author Response

Dear Editor, all,

Thanks to you and the reviewers for your care in reading and making suggestions on our manuscript. The reviewer comments motivated some changes to the paper, which we summarize below; along with detailed notes addressing reviewer comments provided in separate responses.

Thanks again, and best wishes,

Juan and Mike

---

## Round 2 · List of Changes

We have removed the “Gaussian” dataset entirely from the paper, focusing only on tests of the method with the SUSY dataset for now. We have also added comparisons of the CABIN approach to the TMVA kCuts algorithm, and dropped the comparison to the feed-forward, fully-connected neural network in favor of a BDT, which is more commonly used in this context.

With the addition of the BDT and the kCuts comparisons, we felt it better to focus on the comparisons of CABIN with those methods, evaluated in the form of summary ROC curves and cut evolution, rather than showing ROC curves based on CABIN output scores for individual efficiency targets (which were included in the original draft). This has simplified the discussion of the results and focused the attention on the results of CABIN’s regularization procedure.

Finally, we have taken this opportunity to do some language editing, cleanup of the bibliography, and adding some references. In particular, we have now included citations of recent LHC papers that use selection cuts, including a search for SUSY that motivated the use of the SUSY dataset under study.

---

## Editorial Decision

published